# Computer-Aided Design of Boron Nitride-Based Membranes with Armchair and Zigzag Nanopores for Efficient Water Desalination

**DOI:** 10.3390/ma13225256

**Published:** 2020-11-20

**Authors:** Alexey A. Tsukanov, Evgeny V. Shilko

**Affiliations:** 1Center for Computational and Data-Intensive Science and Engineering (CDISE), Skolkovo Institute of Science and Technology (Skoltech), 30, bld. 1, Bolshoy Boulevard, Moscow 121205, Russia; 2Institute of Strength Physics and Materials Science of Siberian Branch Russian Academy of Sciences (ISPMS SB RAS), 2/4, pr. Akademicheskii, Tomsk 634055, Russia

**Keywords:** nanoporous membrane, slit-shaped nanopore, hexagonal boron nitride, desalination, separation, ion rejection, molecular dynamics, computer-aided design

## Abstract

Recent studies have shown that the use of membranes based on artificial nanoporous materials can be effective for desalination and decontamination of water, separation of ions and gases as well as for solutions to other related problems. Before the expensive stages of synthesis and experimental testing, the search of the optimal dimensions and geometry of nanopores for the water desalination membranes can be done using computer-aided design. In the present study, we propose and examine the assumption that rectangular nanopores with a high aspect ratio would demonstrate excellent properties in terms of water permeation rate and ion rejection. Using the non-equilibrium molecular dynamic simulations, the properties of promising hexagonal boron nitride (h-BN) membranes with rectangular nanopores were predicted. It has been found that not only the nanopore width but also its design (“armchair” or “zigzag”) determines the permeability and ion selectivity of the h-BN-based membrane. The results show that membranes with a zigzag-like design of nanopores of ~6.5 Å width and the armchair-like nanopores of ~7.5 Å width possess better efficiency compared with other considered geometries. Moreover, the estimated efficiency of these membranes is higher than that of any commercial membranes and many other previously studied single-layer model membranes with other designs of the nanopores.

## 1. Introduction

The lack of fresh and potable water is a major challenge for many regions in the world. According to the World Health Organization and UNICEF (2019), about 2.2 billion people in the world lack access to safely managed drinking water services [1]. Freshwater is also essential for agricultural and medical needs, as well as for various industrial cycles, including mining. In many cases, the content in water not only of impurity salt ions but also of other common ions should be close to zero. One of the ways to obtain pure freshwater is desalination of salt (in particular, sea) water. The need for high-quality selective filtration of large volumes of water is among the key problems in this field. The most important requirements for materials for selective filtration are a high degree of rejection of impurity ions and molecules, minimal resistance to water flow, high energy efficiency, and the capability of scaling the filtering system to the scale of practical (including industrial) application [2,3,4].

Recent experimental and theoretical investigations have shown that a solution to the problems related to the effective water desalination and decontamination, ions and gases separation should be sought in the field of nanotechnology by the creation of membranes based on the artificial nanoporous materials [5,6,7,8]. Desalination membranes designed on the basis of atomically thin two-dimensional nanomaterials have attracted many researchers’ attention due to predicted superior water desalination efficiency [9,10].

Ion rejection and water permeation are two key performance indicators in water desalination. These two indicators are connected, as a rule, oppositely: an increase in one indicator is accompanied by a decrease in another one. Modern membranes, which are widely used for water purification and desalination, are usually characterized by either high salt rejection (filtration selectivity) and relatively low water permeability, or an insufficient level of salt retention, but high water permeability [11]. It is important to note that an increase in the specific area of pores in the section of the membrane reduces the stability of the material and its strength under conditions of high applied working pressure [12]. Therefore, the development of efficient nanoporous atomically thin membranes (NATM) for desalination and water purification is both a search for a compromise between the abovementioned indicators and a search for new materials that are more stable (strong), as well as a search for an effective geometry of nanopores and chemical structure of nanopore surface or rim. Such a problem can be efficiently solved using computer-aided design and atomistic numerical modeling, prior to the stage of expensive synthesis and experimental testing.

Various promising NATMs with nanopores characterized by various geometry and electrostatic properties of surface or perimeter have been proposed and theoretically investigated in the last decade.

The potential of nanoporous graphene as a two-dimensional material for efficient water desalination was analyzed by Cohen-Tanugi et al. [13] The authors considered the pores in the form of a rounded hexagon with 12 hydrogen (H) atoms (hydrogenated pore), as well as with 6 H atoms and 6 OH-groups (hydroxylated pore) arranged in an alternating manner on the perimeter. The results of a theoretical study showed that graphene with a hydroxylated pore has a slightly higher permeability compared to a hydrogenated pore. In both cases, salt rejection is ~100%, and productivity is ~39–66 L/(cm^2^·day·MPa). The value of the latter indicator is ~2 orders of magnitude higher than that of commercial reverse osmosis membranes [14].

Nanoporous boron nitride nanosheets with different pore geometry (rhombus, circle) and different chemical groups (-H, -F, -OH) on the pore perimeter were studied numerically in [15]. It was shown that a nanopore with 6 OH–groups and 6 -BH on the pore edge shows the best efficiency in water desalination: productivity ~29.27 L/(cm^2^·day·MPa) and about 95% salt rejection.

Nanopores in promising membranes based on two-dimensional covalent triazine frameworks functionalized with chlorine (CTF-1-Cl) or methyl groups (CTF-1-CH3) have the shape of a six-pointed star. These membranes are computationally studied in [16]. In the first case (CTF-1-Cl), the pore perimeter consists of six negatively charged chlorine atoms, and in the second case (CTF-1-CH3), it consists of 12 hydrogen atoms with a small positive partial charge. The predicted productivity of CTF-1-Cl is 33.3 L/(cm^2^·day·MPa) at ~96% salt rejection, while CTF-1-CH3 has productivity 24.6 L/(cm^2^·day·MPa) and ~100% salt rejection.

Promising desalination membranes based on a boron nitride nanosheet with triangular nanopores were computationally studied in [17]. The perimeter of nanopores was formed only by nitrogen atoms (partial charge −0.14–0.16 e) or only by boron atoms (partial charge 0.11–0.12 e). Modeled functional conditions included mechanical stretching. Simulation results showed that nanopores with a perimeter formed by nitrogen atoms are able to achieve 100% ion rejection and productivity 8.64-11.41 L/(cm^2^·day·MPa). Triangular nanopores in multilayer graphitic carbon nitride were considered in [18]. The authors showed that multilayer graphitic carbon nitride structures are capable of rejecting 100% of Cl^−^ and Ca^2+^ ions, however, the rejection of Na^+^ ions is estimated at ~80%.

Nanopores in a synthesized two-dimensional conjugated aromatic polymer (2D-CAP) have a rectangular net section [19]. The pore perimeter is formed by 14 hydrogen atoms and 4 nitrogen atoms, which are located somewhat deeper from the edge of the nanopore. According to quantum mechanical calculations, nitrogen atoms have a negative partial charge -0.445 e, and hydrogen atoms have a small positive charge of about 0.15 e [20]. Numerical estimates of the filtration characteristics of 2D-CAP membranes predict close to 100% salt retention and high productivity of ~28 L/(cm^2^·day·MPa).

Despite the wide variety of computational studies of nanoporous materials with ~1 nm nanopores as promising materials for desalination and water purification, membranes with slit-shaped nanopores remain outside the focus of research. Hereinafter, the term “slit-shaped nanopore” implies rectangular nanopores with a high aspect ratio (note that although the pores in the abovementioned 2D-CAP membrane have a rectangular cross-section, their aspect ratio is low; this does not allow us to classify them as slit-shaped). In this regard, only a few recent works should be mentioned. Yamada et al. studied water filtration through the slit-shaped nanopore in graphene [21,22]. However, they considered the flow of water through slits of arbitrary width, which is not realized in the case when nanopores are in a nanosheet with a certain crystal structure. Moreover, the filtration of impurity ions was not studied in this work. We should also note the results of another study, indicating that membranes with slit-shaped pores could have higher performance in comparison with membranes with cylindrical pores, i.e., greater selectivity at a given value of the permeability [23]. The authors of the present paper recently carried out a molecular dynamics study of the selectivity and permeability of slit-shaped nanosized pores in a natural mineral (hydroxyapatite) [24]. We showed that sub-nanometer wide slit-shaped nanopores in hydroxyapatite are capable of both good salt ion rejection and high water permeability.

Based on this background, in the present study, we propose and check the assumption that NATM with rectangular nanopores characterized by a high aspect ratio would demonstrate high enough indices of water permeation rates and impurity ion rejection. We chose the hexagonal boron nitride (h-BN) nanosheet as the basis for the nanoporous atomically thin membrane. The choice of the h-BN nanosheet (BNNS) as the base material is substantiated by several reasons. First, an h-BN nanosheet possesses high mechanical strength [25]. Secondly, there are several techniques to produce nanopores with controlled geometry in BNNS [26,27]. Besides, BNNS can be functionalized by plenty of different chemical groups/ligands [28]. BNNS also exhibits excellent sorption performances for a wide range of pollutants such as oils, organic solvents, and dyes [29,30]. Also note that the nanoporous BNNS is a very promising base for the development of high-spatial-sensitivity nanopore electrical devices for various applications, including DNA detectors [31,32]. Furthermore, h-BN is also interesting due to the naturally present partial charges in the B-N atom-pairs, which, depending on nanopore design, may provide us with an additional factor to control the selectivity of the membrane for charged species. It is worth noting that there are several theoretical works devoted to the investigations of the efficiency of nanoporous BNNS in the separation of gases [33], heavy metal ion removal [34], and DNA sequencing [35].

We carried out the study of water permeability and desalination efficiency of BNNS with slit-shaped nanopores using the non-equilibrium molecular dynamics simulations.

## 2. Materials and Methods

To get the theoretical estimates of water permeability and ion rejection of nanoporous h-BN nanosheet with rectangular pores characterized by high aspect ratio, we conducted the explicit atomistic modeling of saltwater filtration through the fragment of the model membrane. Besides the Na^+^ and Cl^−^ ions, 4 alkali (Li^+^, K^+^, Rb^+^, Cs^+^) and 3 halide ions (F^−^, Br^−^, I^−^) were also considered in the present study as well as widespread divalent cations Mg^2+^ and Ca^2+^.

To identify an optimal nanopore design in terms of permeability rate–ion rejection balance, we developed seven models of h-BN nanosheet with slit-shaped nanopores of different widths (Table 1). Depending on the orientation of the longer side of the pore relative to the principal crystal axis of the h-BN, these models can be divided into two groups: armchair and zigzag pores (Figure 1a). Nanopores of the first group have a longer side of pore rim consisting of (–B-N–)_n_ pairs. The longer side of nanopores, which belong to the second group, consists of only B or only N atoms. Unlike pure (not-functionalized) graphene-based membranes, there are significant partial electrical charges at B and N atoms due to the difference in electronegativity of these elements.

We used the classical MD force field (FF) parameters developed by Blankschtein et al. [36] for the h-BN. According to this FF model, partial atomic charges of B and N atoms are 0.907 e and −0.907 e, respectively. Thus, there is an electric field orthogonal to the larger side of the pore in the case of zigzag nanopores.

The pore effective width *d_eff_* can be approximately calculated using the relation $d_{eff}=d_{c}-\left(r_{B}^{vdw}+r_{N}^{vdw}\right)$, where the width of pore *d_c_* is defined by centers of edge atoms, and $r_{B}^{vdw}=1.8\;Å$ and $r_{N}^{vdw}=1.6\;Å$ are van der Waals radii of boron and nitrogen atoms, respectively [37]. Note that the effective length of the armchair and zigzag nanopores (defined similarly to effective width) is slightly different in our simulations: *L_eff_* = 44.8 Å for armchair nanopore, and *L_eff_* = 41.3 Å for zigzag nanopore.

The modified TIP3P model was used to parameterize the water molecules [38]. Force field parameters for alkali and halide ions were adopted from [39]. Parameterization of cations Ca^2+^ and Mg^2+^ was made in accordance with a CHARMM force field [40]. All these ion models are compatible with the used model of water TIP3P.

The simulation setup is a *y*-*z*-periodic domain with fixed dimensions *y* = 52.056 Å and *z* = 50.091 Å (Figure 1b). The length of the simulation box along the *x*-axis is about 170 Å and varies during simulation depending on the pressure drop in the system. The computational domain includes three h-BN nanosheets, namely L, M, and R, which divide the system into two reservoirs V_1_ (between L and M nanosheets) and V_2_ (between M and R nanosheets, see Figure 1b). Reservoir V_1_ plays the role of the feeder and is filled with water and ions. It initially contains about 13,600 water molecules and 194 ions. Outlet reservoir V_2_ initially contains only water (about 1800 molecules). The equally distributed external force was applied to the L nanosheet along the *x*-direction to provide the pressure *p*_0_ + Δ*p*. In such a way the force was also applied to the R nanosheet to provide pressure *p*_0_ = 0.1 MPa. The pressure drop Δ*p* was varied within the range from 50 to 200 MPa (and 100–500 MPa for Armchair-1). The order of magnitude of Δ*p* is typical for such types of simulations [41,42,43]. 

The ion composition and content in the feeder volume V_1_ depends on the ions examined. Initial concentration in the case of sodium chloride was (NaCl) = 660 mmol/l (660 mM), which is about 10% higher than the average salinity of ocean water (~600 mM). For all salt models, the initial concentrations are presented in Table 2. Note that in all considered cases, the total number of ions in the feeder volume was the same.

All the simulations were conducted using a LAMMPS (version: 30 Jul 2016) software package (Large-scale Atomic/Molecular Massively Parallel Simulator by Sandia National Laboratories, Livermore, CA, USA) [44,45]. The calculations were performed on the Lomonosov-2 supercomputer (Lomonosov Moscow State University, Russia) [46,47] and the supercomputer Zhores (CDISE, Skoltech, Russia) [48].

## 3. Results and Discussion

### 3.1. Water Permeability

The results of the performed non-equilibrium MD simulations of water filtration through the model membranes are presented in Figure 2a. To compare characteristics of individual nanopores depending on their orientation and effective width, we recalculate for single pore the permeabilities obtained for the model membranes, taking into account the number of slits in their working cross-section Figure 2b.

As we can expect, the nanopore Armchair-1 (*d_c_* = 5.01 Å) is impermeable for water at least in pressure drop Δ*p* range up to 500 MPa due to the steric factor. Indeed, the mean van der Waals diameter of the water molecule is ~2.82 Å [49], whereas the effective width of the nanopore is only ~1.61 Å, which is significantly less than the characteristic size of a water molecule (Table 1).

The number of water molecules filtered through a single nanopore of each type as a function of time is shown in Figure 2b. Since a higher number of narrow nanopores can be accommodated in a unit running length of a membrane than wide ones, the results for individual nanopores (Figure 2b) look different than for membranes with a system of nanopores (Figure 1a and Figure 2a). The results recalculated for the individual nanopores made it possible to construct the following sequence of nanopores with increasing permeability: 0 = A1 < A2 < Z1 < A3 < A4 < Z2 < A5 (where “A” means armchair pore and “Z” means zigzag pore). It is interesting to note that the Zigzag-2 (Z2) nanopore has a higher permeability than the Armchair-4 (A4) nanopore, although the effective width of the latter is ~0.1 Å larger. 

It can be seen from the obtained curves (Figure 2a,b) that during at least the first 12 ns of filtration, the time dependence of the number of filtered water molecules is approximately linear for all considered cases. Using the least-squares method, we determined the productivity of the model membranes in conventional units L/(cm^2^·day·MPa), as well as in L/(m^2^·h·bar). The predicted productivity of the considered membranes ranges from 10.2 L/(cm^2^·day·MPa) (for Armchair-2) to 46.8 L/(cm^2^·day·MPa) (for Zigzag-2). These values are comparable with the predicted productivity of the most productive promising nanotechnological membranes [13,15,16,20] and ~2 orders of magnitude higher than the productivity of modern commercial facilities [14]. The estimated performance for each of the considered model membranes in order of increasing productivity is shown in the diagram in Figure 3. 

### 3.2. Salt Rejection

The main performance indicator of water desalination membranes is salt rejection. By recording the number of ions traversing the membrane as well as the number of filtered water molecules, a percentage of salt rejection can be estimated using the following common relation:
(1)$$R\left(\%\right)\;=\;\left(1-\frac{1}{C_{0}}\frac{N_{i}}{N_{w}}\right)\cdot 100\%,$$
where *C*_0_ is an initial ion concentration in the feeder reservoir (V_1_), *N_i_* is the number of ions passed through the membrane, *N_w_* is the number of filtered water molecules.

One can see from Figure 4 that the traditional statement “the higher the permeability of the membrane, the lower the ion rejection” is also true for the considered model membranes. Besides, ion rejection decreases with increasing pressure drop Δ*p*, which is typical for all filtration NATMs, since ion overcoming the energy barrier is facilitated with increasing external hydraulic pressure. Note that the considered range of pressure drop values is significantly higher than typical values in operating desalination systems (up to 10 MPa). This fact gives a reason to state that ion rejection at typical working pressures will be no less than the estimates given here. Thus, the obtained ion rejection values can be considered as a lower estimate.

The plot in Figure 4b shows that the Armchair-3, Zigzag-1, and Armchair-2 membranes have ~100% salt rejection even at anomalously high values of differential pressure (100 MPa). Therefore, at pressures < 10 MPa, the expected result for these membranes in terms of NaCl salt rejection will also be ~100%.

Divalent cations are responsible for incrustation or deposits on water treatment equipment, which is one of the problems accompanying water processes and recycling [50,51]. Therefore, we carried out additional studies of Ca^2+^ and Mg^2+^ ions rejection at pressure drop Δ*p* = 100 MPa. Taking into account the above results for salt (Figure 4b), we limited ourselves to the study of membranes Armchair-3 and Zigzag-1. The diagram in Figure 5a shows that both membranes completely reject divalent cations. This also means that at a lower operating pressure, the result will be the same. Note that 100% rejection of Ca^2+^ and Mg^2+^ at typical operating pressures can be expected from Armchair-2 membrane, which has a smaller effective slit width.

Although the effective width of nanopores in the Armchair-3 membrane is larger than in the Zigzag-1 membrane, the latter is characterized by a lower value of the Na^+^ ions rejection index. Moreover, the Armchair-5 membrane with a slit width of ~10 Å rejects Na^+^ and Cl^−^ ions by 20% better than a Zigzag-2 membrane with a much smaller slit width (~8.7 Å). This is a consequence of the difference in the structure of the nanopore perimeter and its electrostatic properties. The “mechanism” of the passage (leakage) of the cation Na^+^ through zigzag-like nanopores can be understood by analyzing the MD trajectories of the Na^+^ ions near the pore (Figure 5b) shows the example for Zigzag-1 nanopore). Interaction of the Na^+^ cation with negatively charged nitrogen atoms on the zigzag N-edge of the nanopore side provides “rolling over” to the other side of the membrane near the pore edge.

Such leakage can be prevented by chemical functionalization of the nanopore perimeter. In the simplest version, the pore can be hydrogenated, since a partially de-screened proton, which has a local positive electric charge, should prevent the contact of the Na^+^ cation with membrane nitrogen. Another way is to use a double h-BN nanosheet membrane, where the charges on one layer will be compensated by the opposite charges on the parallel layer. When using a multilayer h-BN, the mechanical strength of the membrane will multiply. This makes it possible to increase the number of slits in the section. For example, a fifth slit can be added to the considered fragment of the Armchair-3 model membrane (Figure 1b), and permeability will increase by 25%. Verification of the effectiveness of these options is the subject of further research.

### 3.3. Decontamination from Alkali and Halide Ions

Since alkali ions Li^+^, K^+^, Rb^+^, Cs^+^, and halide ions F^−^, Br^−^, I^−^ have different ion radii and hydration shell sizes, it is important to study their filtration through slit-shaped nanopores in h-BN and evaluate ion rejection for understanding the influence of size-related factors. Following the same procedure as in the case of NaCl, we estimated ion rejection for Armchair-3 and Zigzag-1 membranes with respect to light ions Li^+^, K^+^, F^−^, as well as ion rejection of Armchair-3, -4 membranes and Zigzag-1, -2 with respect to heavy ions Rb^+^, Cs^+^, Br^−^, I^−^ (Figure 6 and Figure 7).

One can see that the model membranes Armchair-3 and Zigzag-1 reject 100% of potassium, rubidium, and bromine ions. Membranes with Armchair-3, Armchair-4, and Zigzag-1 pore geometries reject 100% of cesium and iodine ions. Armchair-2 will also completely reject Li^+^, K^+^, Rb^+^, Cs^+^, Br^−^, and I^−^ ions. It is interesting to note that fluoride is not completely rejected by any of the considered membranes (Armchair-2 was not considered in a pair with F^−^). Rejection of lithium is low in the case of the Zigzag-1 membrane. Relatively low fluoride and lithium ions removal in comparison to other ions can be explained by the fact that ionic radii of F^−^ and Li^+^ are the smallest among halide and alkali ions. We also note that the Zigzag-2 membrane is not able to efficiently reject even large ions such as cesium and iodine (R~80%).

## 4. Conclusions

Using nonequilibrium molecular dynamics, we analyzed the prospects of NATM with a slit/rectangular nanopore design as a new 2D material for advanced desalination and water purification systems. We showed that nanoporous sheets of boron nitride with rectangular (slit-shaped) pores are a promising basis for water desalination and separation of halogen and alkali metal ions. The most optimal ratio “productivity—ion rejection” is possessed by h-BN with slit-shaped nanopores ~0.65–0.75 nm wide. The lower estimate of the productivity of such membranes is 19.9–26.4 L/(cm^2^·day·MPa) at >97% rejection of Na^+^, K^+^, Cl^−^, Br^−^, I^−^, Rb^+^, Cs^+^, Mg^2+^, Ca^2+^ ions. This allows considering these membranes among the most efficient filtration nanomaterials. The Armchair-4 membrane with nanopores ~0.88 nm wide rejects 100% of heavy Cs^+^ and I^−^ ions, and has a predicted productivity of ~34.4 L/(cm^2^·day·MPa).

It is worth noting that the nanopores in the hexagonal BN nanosheet can be chemically functionalized. This can significantly increase the selectivity and efficiency of the considered type of NATM. The selection of the most optimal functional groups is a promising task in computer design. It also seems promising to evaluate the efficiency of multilayered h-BN with slit-shaped nanopores. The computational solution of these tasks is planned for the near future.

The presented theoretical estimates of water permeability and impurity ion rejection were obtained under ideal conditions, when NATM is free from defects and not deformed (positions of atoms of the membrane are fixed). Nevertheless, these estimates can be compared with known theoretically or numerically obtained estimates for other promising nanoporous materials under similar idealized conditions. The comparison shows that the efficiency of NATM based on a h-BN nanosheet with rectangular nanopores (even without optimization of the chemical composition of pore perimeter) is one of the highest among promising NATMs with close to 100% ion rejection. In particular, membrane of type Zigzag-1 has a predicted (theoretical) capacity of 26.4 L/(cm^2^·day·MPa) and >98% salt rejection. These indicators are slightly inferior only to Graphyne-3 and nanoporous graphene with a productivity of 29.3 L/(cm^2^·day·MPa) and 39-66 L/(cm^2^·day·MPa), respectively [10,13,52].

It is quite possible that some of the nanoporous membranes currently being developed using computer-aided design (including those discussed in this article) will become the main element in the supply of fresh and drinking water for planned missions to Mars, Callisto, and Enceladus in the near future.

## Figures and Tables

**Figure 1 materials-13-05256-f001:**
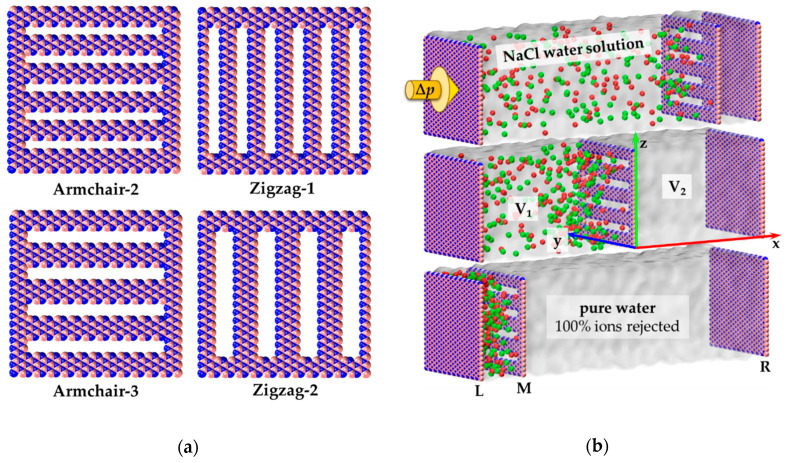
Models of nanoporous membranes and the simulation setup: (**a**) Armchair and zigzag design of nanopores in h-BN nanosheet; (**b**) Simulation domain comprises two volumes—the feeder filled with NaCl water solution (left) and the outlet reservoir (right). The model contains three h-BN nanosheets L, M, and R. The middle one (M) is a nanoporous membrane, and its coordinates are fixed. Plane of M nanosheet has a position *x* = 0, axis *x* is oriented in the direction of the filtration. Axes *y* and *z* are oriented parallel to the armchair and zigzag edges of h-BN, respectively. Extra pressure Δ*p* is applied to nanosheet L (yellow arrow). Atom colors: boron—pink, nitrogen—blue, sodium—red, chlorine—green, water oxygen, and hydrogen atoms are not shown for clarity.

**Figure 2 materials-13-05256-f002:**
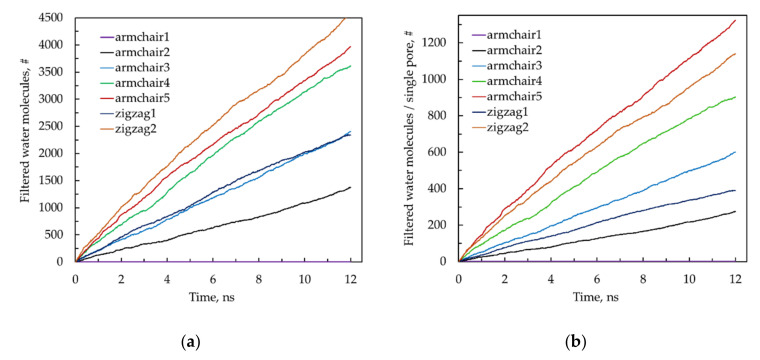
Number of filtered water molecules through the periodic fragment of the model membrane (**a**) and a single (individual) nanopore (**b**) as a function of time. See also Appendix A (MD simulation of water desalination with Armchair-3 model membrane at Δ*p* = 50 MPa).

**Figure 3 materials-13-05256-f003:**
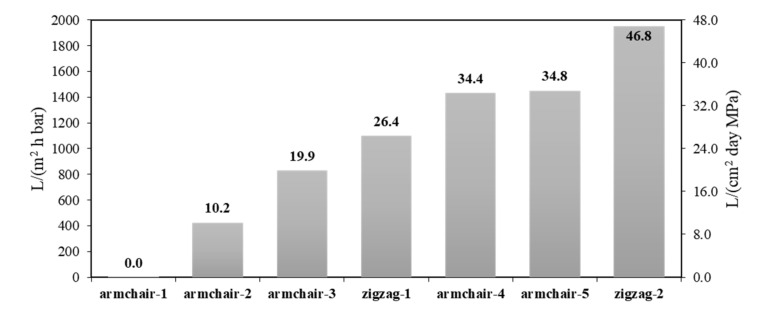
The productivity of model membranes. The numbers above the bars show values in L/(cm^2^ day MPa).

**Figure 4 materials-13-05256-f004:**
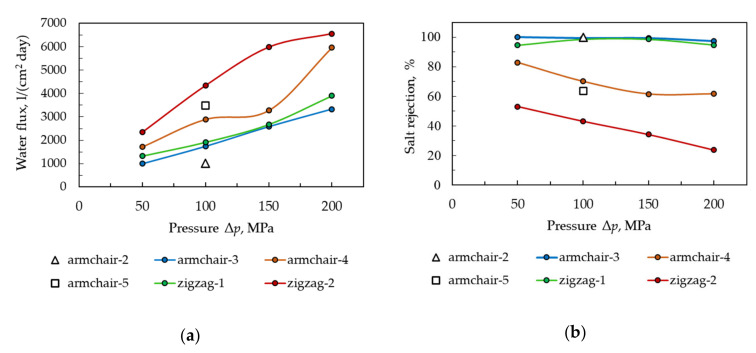
Water flux (**a**) and the percentage of salt rejection (**b**) of model membranes as a function of applied pressure drop.

**Figure 5 materials-13-05256-f005:**
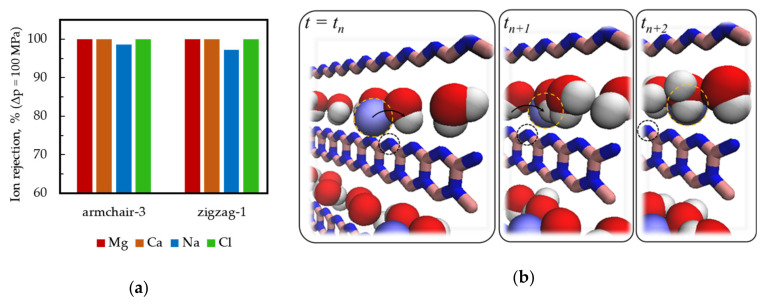
(**a**) NaCl salt and divalent cations rejection rates for Armchair-3 and Zigzag-1 membranes (pressure drop Δp = 100 MPa); (**b**) An explanation of sodium leakage through the rectangular nanopore of zigzag type. The cation (blue sphere with yellow contour) interacts with the negatively charged side of the pore and “rolls” over the nitrogen atom (highlighted by a black circle) at the N-zigzag side. Colors: Na^+^—light-blue, O—red, H—white, B—pink, N—blue. Water molecules are not shown.

**Figure 6 materials-13-05256-f006:**
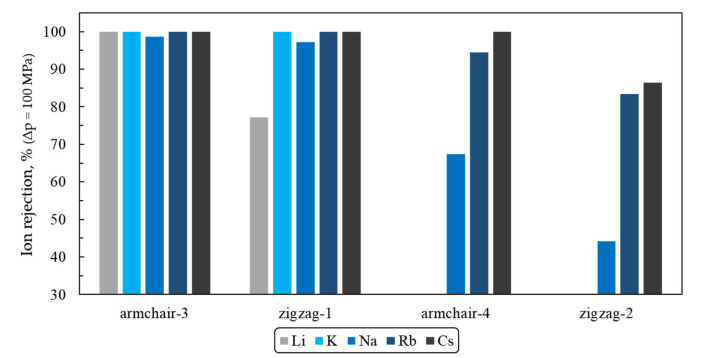
Estimates of cation rejection (pressure drop Δp = 100 MPa). The determination of Li^+^ and K^+^ ions rejection was not carried out for Armchair-4 and Zigzag-2 membranes.

**Figure 7 materials-13-05256-f007:**
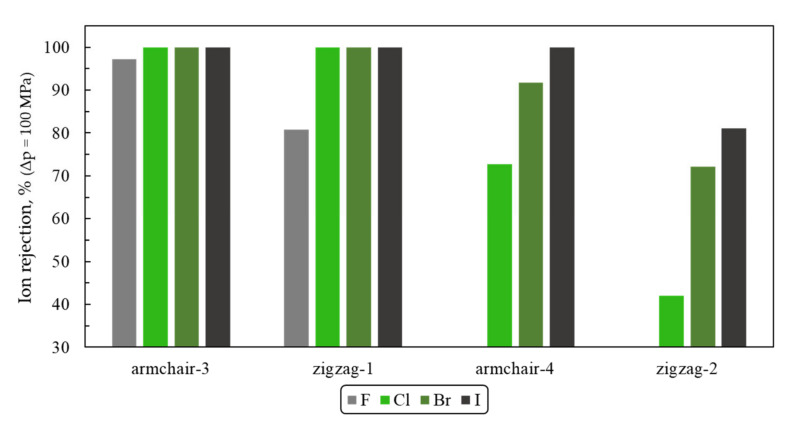
Halide-ions rejection at pressure drop Δp = 100 MPa (a rejection rate of Armchair-4 and Zigzag-2 membranes for fluorine anions F^–^ was not assessed).

**Table 1 materials-13-05256-t001:** Design of the model nanoporous membranes considered in the study.

Model Name	Width *d_c_*, Å	Effective Width *d_eff_*, Å	# of Pores ^1^
Armchair-1	5.01	1.61	6
Armchair-2	6.262	2.86	5
Armchair-3	7.514	4.11	4
Armchair-4	8.766	5.37	4
Armchair-5	10.018	6.62	3
Zigzag-1	6.507	3.11	6
Zigzag-2	8.676	5.28	4

^1^ Number of pores in the periodic fragment of the membrane.

**Table 2 materials-13-05256-t002:** Initial concentrations of ions in the feeder volume.

	Model Salt	Salt Concentration, mM	Ion	Ion Concentration, mM
1.	NaCl	660	Na^+^	660
			Cl^−^	660
2.	CaCl_2_	440	Ca^2+^	440
			Cl^−^	880
3.	MgCl_2_	440	Mg^2+^	440
			Cl^−^	880
4.	(Li + K) (F + Br)	330	Li^+^	330
			K^+^	330
			F^−^	330
			Br^−^	330
5.	(Rb + Cs) (Br + I)	330	Rb^+^	330
			Cs^+^	330
			Br^−^	330
			I^−^	330

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
