# Peer review of "Computer-Aided Design of Boron Nitride-Based Membranes with Armchair and Zigzag Nanopores for Efficient Water Desalination"

_materials, 2020, doi:10.3390/ma13225256_

Round 1

Reviewer 1 Report

Topic described in paper is interesting and worth to develop. 

In spite of all there are few remarks:

Line 181: ,,The ion composition and content in the feeder volume V1 depend on ions examined. Initial concentrations were: [NaCl] = 660 mM, [CaCl2] = 440 mM, [MgCl2] = 440 mM, and[(Li+K)(F+Br)] = 330 mM or [(Rb+Cs)(Br+I)] = mM. Note that the salinity of water in the model is
10% higher than average salinity of ocean water"

I think it will be better to put data about concentrations of all ions in a table (it will be easier to read)

Fig.6 - Why Li removal is so low in comparison to other elements?
Fig.7 - Why fluoride removal has such low value?

Was vulnerability to fouling examined?

Author Response

We thank the reviewer for valuable comments. We attentively studied all comments and made corresponding changes in the manuscript. The changes in the revised manuscript are highlighted by yellow.

Below is the point by point response to the reviewer’s comments and suggestions.

Sincerely,

The authors

Reviewer 1

Topic described in paper is interesting and worth to develop. 

In spite of all there are few remarks:

Comment:

Line 181: "The ion composition and content in the feeder volume V1 depend on ions examined. Initial concentrations were: [NaCl] = 660 mM, [CaCl2] = 440 mM, [MgCl2] = 440 mM, and[(Li+K)(F+Br)] = 330 mM or [(Rb+Cs)(Br+I)] = mM. Note that the salinity of water in the model is 10% higher than average salinity of ocean water".

I think it will be better to put data about concentrations of all ions in a table (it will be easier to read)

Response:

We agree with this comment and have put data about initial ions concentrations in Table 2 (revised manuscript).

Comment:

Fig.6 – Why Li removal is so low in comparison to other elements?
Fig.7 - Why fluoride removal has such low value?

Response:

Relatively low fluoride and lithium ions removal in comparison to other ions can be explained by the fact that ionic radii of F and Li+ are the smallest among halide and alkali ions. We have added this explanation in the manuscript; see lines 309-311 of the revised manuscript.

Comment:

Was vulnerability to fouling examined?

Response:

Unfortunately, the duration of the simulation is insufficient to comprehensively examine the fouling process due to the very small value of the time step of the integration scheme. Such a study is a future step in our research activity in this field. It is planned on larger models.

Reviewer 2 Report

In this manuscript the authors studied water desalination performance of atomically thin hexagonal boron nitride (h-BN) membranes possessing rectangular nanopores by means of the computer-aided design. As I am not in the field of simulation, I have several basic questions regarding how the simulation works.

  1. How did you choose the values of x, y, and z in Figure 1(b)? Were they arbitrarily selected?
  2. Based on the video S1, the position of nanosheet M is fixed, while nanosheet L and R move toward x-direction at the same speed, and hence V1 an V2 change at the same rate. How did you determine the rate of volume change? Is such rate determined based on the rate of water permeation through membrane M?

Author Response

We thank the reviewer for valuable comments. We attentively studied all comments and made corresponding changes in the manuscript. The changes in the revised manuscript are highlighted by yellow.

Below is the point by point response to the reviewer’s comments and suggestions.

Sincerely,

The authors

Reviewer 2

In this manuscript, the authors studied water desalination performance of atomically thin hexagonal boron nitride (h-BN) membranes possessing rectangular nanopores by means of the computer-aided design. As I am not in the field of simulation, I have several basic questions regarding how the simulation works.

Comment:

How did you choose the values of x, y, and z in Figure 1(b)? Were they arbitrarily selected?

Response:

Axis x,y,z and origin of the coordinate system were chosen for convenience: filtration direction is along x, the position of the plane of the model membrane is x = 0. Axles y and z are in the plane of the membrane. They are oriented to be parallel to the armchair and zigzag edges of h-BN, respectively. We have added these details to the Figure 1 caption.

Comment:

Based on the video S1, the position of nanosheet M is fixed, while nanosheet L and R move toward x-direction at the same speed, and hence V1 an V2 change at the same rate. How did you determine the rate of volume change? Is such rate determined based on the rate of water permeation through membrane M?

Response:

We did not control the velocities of pistons. Instead, we controlled the pressure (p0+∆p for L piston, and p0 for R piston) by applying the external forces as described in methods (lines 179-184) of the revised manuscript. We think that this way better corresponds to the real situations where the solution is filtered by the differential pressure. Moreover, this scheme is typical for such types of simulations (for example, see the references [41-43]).
